# Phytochemicals Targeting JAK–STAT Pathways in Inflammatory Bowel Disease: Insights from Animal Models

**DOI:** 10.3390/molecules26092824

**Published:** 2021-05-10

**Authors:** Sun Young Moon, Kwang Dong Kim, Jiyun Yoo, Jeong-Hyung Lee, Cheol Hwangbo

**Affiliations:** 1Division of Applied Life Science (BK21), PMBBRC and Research Institute of Life Sciences, Gyeongsang National University, Jinju 52828, Korea; symoon0414@gnu.ac.kr (S.Y.M.); kdkim88@gnu.ac.kr (K.D.K.); yooj@gnu.ac.kr (J.Y.); 2Division of Life Science, College of Natural Sciences, Gyeongsang National University, Jinju 52828, Korea; 3Department of Biochemistry, College of Natural Sciences, Kangwon National University, Chuncheon 24341, Korea; jhlee36@kangwon.ac.kr

**Keywords:** inflammatory bowel disease (IBD), janus kinase (JAK), signal transducer and activator of transcription (STAT), phytochemicals

## Abstract

Inflammatory bowel disease (IBD) is a chronic inflammatory disorder of the gastrointestinal tract that consists of Crohn’s disease (CD) and ulcerative colitis (UC). Cytokines are thought to be key mediators of inflammation-mediated pathological processes of IBD. These cytokines play a crucial role through the Janus kinase (JAK) and signal transducer and activator of transcription (STAT) signaling pathways. Several small molecules inhibiting JAK have been used in clinical trials, and one of them has been approved for IBD treatment. Many anti-inflammatory phytochemicals have been shown to have potential as new drugs for IBD treatment. This review describes the significance of the JAK–STAT pathway as a current therapeutic target for IBD and discusses the recent findings that phytochemicals can ameliorate disease symptoms by affecting the JAK–STAT pathway in vivo in IBD disease models. Thus, we suggest that phytochemicals modulating JAK–STAT pathways are potential candidates for developing new therapeutic drugs, alternative medicines, and nutraceutical agents for the treatment of IBD.

## 1. Introduction

Plants’ phytochemicals have been used as a source of traditional medicine for millennia [1]. A large portion of current drugs for disease treatment have originated from plants, even though new drugs have been developed using synthetic chemistry [2]. Recently, the significance of phytochemicals has been emphasized for therapeutic applications with fewer side effects in various inflammation-related diseases, including cancer, diabetes, rheumatoid arthritis, and inflammatory bowel disease (IBD) [3,4]. Phytochemicals are good sources of new anti-inflammatory drugs that regulate various inflammatory responses against inflammatory diseases [5].

Inflammatory bowel disease is a typical chronic inflammation-mediated disease of the gastrointestinal (GI) tract [6]. IBD is typically classified into two major forms of chronic and relapsing-remitting inflammation: Crohn’s disease (CD) and ulcerative colitis (UC), which is increasing in incidence and prevalence worldwide [7]. Although both diseases have common clinical features, such as diarrhea and abdominal pain, they differ in various aspects. Crohn’s disease is characterized by discontinuous inflammation at any location in the digestive tract, from the mouth to the anus, which can spread into the deeper layers of the bowel [8]. On the other hand, ulcerative colitis is a long-standing chronic inflammation of the colonic mucosa, which affects certain parts or the entire colon, and is most commonly limited to the mucosal surface [9]. Despite the fact that CD and UC are distinct entities, the precise causes of disease pathogenesis remain largely unknown.

The significance of treatment and prevention of IBD has been steadily increasing [10]. Although many drugs have been developed to treat IBD, these drugs have adverse effects on the GI tract [11]. In addition to conventional drugs, there have been important advances in IBD therapy targeting cytokines that have been well documented to play a key role in both the chronic and relapsing phases of IBD [12]. For example, tumor necrosis factor (TNF) inhibitors have been progressively used as therapies for IBD. However, the development of new drugs has been emphasized because unresponsive patients or patients who have lost response to anti-TNF therapy are growing, and a wide array of cytokines besides TNF are involved in the pathogenesis of IBD [13,14,15]. Currently, inhibitors of JAK and STAT that prevent multiple pro-inflammatory cytokine signaling pathways in IBD have been considered as new therapeutic approaches [16,17]. Most cytokines in IBD play crucial roles in chronic inflammatory responses by activating the JAK–STAT pathways (Figure 1) [18]. Binding of cytokines to their respective transmembrane receptors promotes the activation of JAK, which allows the translocation of STATs to the nucleus. Eventually, they regulate the transcription of specific target genes [19]. Accumulating evidence supports the idea that therapeutic intervention of the JAK–STAT pathway can efficiently modulate the complex inflammation driven by various cytokines in IBD [20]. In particular, tofacitinib, a JAK inhibitor, has already been approved and is clinically used for UC patients, and many other inhibitors are now being studied in preclinical and clinical trial phases [17]. JAK inhibitors are small molecules that block JAK activity, which transduces signals from cytokine-receptors to STAT [21]. In addition, direct inhibitors of STAT have been studied in preclinical models of IBD, even though no clinical trials have been undertaken for IBD patients [22]. Compared to conventional treatments, targeting the JAK–STAT pathway is considered a new therapeutic strategy for IBD patients [20].

Anti-inflammatory phytochemicals are known to have anti-IBD activity by modulating the production of inflammatory cytokines [23]. To investigate the function of phytochemicals targeting JAK–STAT pathways in IBD, we searched for articles in PubMed with three main key words (phytochemical, IBD, and anti-inflammation). In addition, we limited phytochemicals that regulated JAK–STAT pathways in in vivo animal model systems. Numerous studies have shown that phytochemicals, including phenolic compounds, terpenoids, alkaloids, and organosulfur compounds, could be used as therapeutic agents as they modulate cellular inflammatory mechanisms associated with IBD [24]. In particular, some phytochemicals have been reported to inhibit activation of the JAK–STAT pathway [25]. Considering the advantages of phytochemicals with few side effects, phytochemicals targeting the JAK–STAT pathways would be a good source of new drugs for the treatment of IBD [26].

This review describes the current understanding of the JAK–STAT pathway in IBD and highlights the significance of the JAK–STAT pathway as a therapeutic target for IBD. In addition, we discuss the evidence that phytochemicals induce IBD remission by affecting the JAK–STAT pathway in animal models of IBD. Furthermore, we analyzed human-relevant equivalent doses of each phytochemical modulating the JAK–STAT pathway and suggest that phytochemicals have strong potential for the development of new anti-JAK–STAT drugs for IBD.

## 2. JAK–STAT Signaling Pathway in IBD

The JAK–STAT pathway is a well-conserved signaling pathway and is involved in many cellular processes, including cell division, cell death, and regulatory immune function [27]. The JAK–STAT pathway plays a pathogenic role in many diseases, and its hyperactivation is associated with inflammatory and autoimmune diseases such as rheumatoid arthritis, IBD, systemic lupus erythematosus, and psoriasis [28]. Although the etiology of IBD remains unknown, mucosal immune and non-immune cells in the inflamed gut of IBD patients spontaneously release pro-inflammatory cytokines such as TNF-α, interferon gamma (IFN-γ), interleukin (IL) 1 beta (IL-1β), IL-6, IL-8, and IL-12, which play a central pathologic role in IBD [29]. These extracellular cytokines in IBD modulate inflammatory responses by activating the JAK–STAT pathway. The binding of cytokines to their cognate receptors triggers the conformational change of the receptors that alter the position of the associated JAK, resulting in phosphorylation of JAK and tyrosine residues on cytokine receptors [30]. Phosphorylated tyrosine residues on cytokine receptors serve as binding sites for STATs, and the recruitment of STAT to the receptor induces the phosphorylation of STAT by JAK [31]. Ultimately, phosphorylated STATs dissociate from their receptor docking sites and form homo- or heterodimers. After that, they translocate from the cytoplasm into the nucleus, where they regulate the transcription of cytokine-responsive genes (Figure 1) [19]. The JAK–STAT pathway is a significant intracellular downstream signaling mediator used by various inflammatory cytokines that are increased in IBD. Thus, inhibitors targeting the JAK–STAT pathway have the advantage of suppressing multiple cytokine pathways in the treatment of IBD.

### 2.1. JAK Family of Proteins and JAK Inhibitors

The Janus kinase (JAK) protein family is a non-receptor protein tyrosine kinase family that includes four proteins: JAK1, JAK2, JAK3, and tyrosine kinase 2 (TYK2). JAK3 is predominantly found in hematopoietic cells, whereas the expression of JAK1, JAK2, and TYK2 is not restricted to specific tissues [32]. JAK is a critical intracellular signaling mediator that transduces signals from cell surface cytokine receptors to the nucleus in IBD [33]. JAK dysregulation can result in pathological processes in IBD [34]. In addition, various cytokines have been widely accepted as crucial inflammatory mediators leading to immunological events in IBD patients [16]. In this regard, we focus on specific cytokines that have been reported to play a key role in IBD pathogenesis and discuss the JAKs activated by these cytokines.

JAK differentially associates with diverse cytokine receptors activated by various cytokines and activates different types of STAT members [19]. In other words, the JAK protein functions as a transmitter between cytokine receptors and STATs in multiple combinations, which allows the generation of specific responses to many different cytokines [35]. Each JAK protein associates with different subunits of cytokine receptors facilitating multiple combinations with different JAK proteins, which exhibit intracellular complexity of IBD [33,36]. Depending on the activated signaling from specific cytokines to their cognate receptors, the pairing of JAK is determined (Figure 2). Binding of IL-2 family cytokines (IL-2, IL-4, IL-7, IL-9, IL-15, and IL-21) to the type I receptor common γ-chain (γc) activates JAK1 and JAK3 [37,38]. Granulocyte-macrophage colony-stimulating factor (GM-CSF), which has been reported to be increased in the serum of CD patients, binds to the type I receptor β-chain and is mediated through JAK2 [39,40]. IL-6 has been reported to have a direct correlation with disease activity in IBD [41]. Binding of IL-6 to type I receptor common glycoprotein 130 (gp130) primarily activates JAK1 and TYK2, followed by JAK2 and TYK2 [42,43]. IL-12 and IL-23 signal through the IL-12 receptor leads to the activation of JAK2 and TYK2 [44,45]. IL-10 and IL-22 bind to type II cytokine receptors, which activate JAK1 and TYK2 [28]. Signaling between IFN-γ and IFN-γ receptor requires JAK1 and JAK2 [46].

Various inflammatory cytokines are linked to the pathological effects of IBD through JAK proteins [47]. Thus, therapy inhibiting JAK may block multiple proinflammatory cytokine signaling pathways in IBD, unlike therapy targeting cytokines or cytokine receptors. Currently, several JAK inhibitors are being evaluated for the treatment of IBD patients, and one of them, tofacitinib, has already been approved for active UC patients [48] (Table 1). Tofacitinib is a strong selective inhibitor of JAK3 and JAK1 and has modest selectivity for JAK2 and TYK2 [49,50]. It can mainly block proinflammatory cytokines (IL-2, IL-4, IL-7, IL-9, IL-15, and IL-21) by inhibiting JAK1/3 and modulating other cytokines that use JAK2 and TYK2. Other JAK inhibitors are being developed in clinical trials, including peficitinib and TD-1473 as pan-JAK inhibitors [51,52], and filgotinib and upadacitinib with selectivity for JAK1 [53,54]. Although these JAK inhibitors target specific JAKs, higher doses could lead to off-target binding or immunosuppressive adverse effects [55]. Therefore, we could alternatively consider phytochemicals, which can modulate JAK pathways in IBD, with fewer side effects than synthetic chemical drugs for the management of adverse processes related to JAK inhibition.

### 2.2. STAT Family of Proteins and STAT Inhibitors

The signal transducer and activator of transcription (STAT) family, which is a critical transcription factor that mediates cytokine-driven signaling, has been actively investigated in IBD pathology [56]. The STAT protein family is composed of seven proteins: STAT1, STAT2, STAT3, STAT4, STAT5A, STAT5B, and STAT6 [57]. Increased expression and activation of STAT1 has been reported in active IBD patients [56,58]. However, the function of STAT1 depends on the cell type in IBD; it is pro-inflammatory in lymphocytes and anti-inflammatory in macrophages/intestinal epithelial cells [59,60]. The phosphorylation of STAT1 is mediated by JAK1/JAK2 or JAK1/TYK2 and has fundamental relevance to signaling via the IFN-γ and related family of receptors [46,61]. STAT1 is also known to be activated by gp130 and γC family receptors [62].

STAT2 is mostly involved in type I interferon (IFN-α and IFN-β) [63]. Although STAT2 has been less studied in IBD pathology compared to other STAT proteins, STAT2 has been suggested to be downregulated in IBD [56].

STAT3 has been well studied to have a fundamental role in IBD. STAT3 is phosphorylated by JAK1, JAK2, or TYK2 activated via signaling of the gp130 family of cytokines (IL-6 and IL-11) or IL-10 family members such as IL-10 and IL-22 [64]. Several studies have reported that the expression and phosphorylation of STAT3 are increased in IBD [65,66]. In addition, downregulation of STAT3 has been shown to improve disease severity in a murine model of colitis [67,68]. The IL-6-STAT3 signaling is involved in the proliferation of lamina propria T cells and blocking of these attenuates chronic intestinal inflammation in experimental colitis [69]. However, similar to STAT1, STAT3 is known to play a role in both pro- and anti-inflammatory effects. STAT3, which is activated by cytokines such as IL-22 and IL-10, plays a protective role in IBD. IL-22 has been reported to induce wound healing, resulting in epithelial regeneration [70,71,72]. STAT3 phosphorylation by IL-10, which is produced in a wide range of innate leukocytes (macrophages, neutrophils, and dendritic cells), might play a role in preventing the disease in experimental colitis models [73,74]. Taken together, STAT3 promotes pro-inflammatory signals in acquired immune cells in IBD, whereas its role in innate immune cells is the suppression of colitis by enhancing mucosal protection.

STAT4 is phosphorylated by JAK2 and TYK2 in response to IL-12- or IL-23-dependent signaling [44,75]. STAT4 is thought to be linked to IBD based on its essential role in the function of T helper type 1 (Th1) cells, which are thought to be important for CD pathogenesis [76]. STAT4 signaling in response to IL-12 is involved in promoting inflammatory reactions by inducing the expression of the Th1-secreted cytokine, IFN-γ [77]. Increased expression of STAT4 in IBD patients has been shown to be involved in chronic inflammation [78,79]. Indeed, STAT4 knock out mice showed protective effects against experimental colitis [80,81]. Thus, targeting STAT4 may have therapeutic potential against IBD.

STAT5 is predominantly activated through JAK1 and JAK3 in response to the γC family of receptors by IL-2, -7, -9, -15, and -21, and is also activated by JAK2 in response to the type I receptor β-chain by the IL-3 family [38,82]. Several studies have shown that STAT5 plays a protective role in colitis. As a protective mechanism, STAT5 has been shown to be essential for the proliferation of intestinal epithelial stem cells, leading to the regeneration of crypt epithelium [83]. STAT5 also plays a crucial role in IL-2 dependent forkhead box P3 (FOXP3) induction in Treg cells that can prevent intestinal inflammation in experimental colitis [84,85]. Thus, STAT5 may not be an appropriate therapeutic target for the treatment of IBD.

STAT6 phosphorylation arises from JAK1 and JAK3, similar to STAT5; however, it is only induced by the γC family of receptors such as IL-4R and IL-13R [62]. STAT6 has been shown to be involved in T helper cell type 2 (Th2)-dependent IBD pathology and to have pro-inflammatory properties via the regulation of Th2 cytokines [86,87]. The phosphorylation of STAT6 was observably increased in the tissue of UC patients [88,89].

As shown above, STAT proteins could be attractive targets for the regulation of intestinal inflammation in addition to JAK for the treatment of IBD. Indeed, direct inhibitors that block STAT proteins have long been studied for treating inflammatory and autoimmune diseases, including IBD [90]. Small molecule compounds inhibiting STAT1 signaling have been shown to improve disease in experimental colitis by selective sequestering of STAT1 from the receptor [91]. STAT3 inhibitors reduce DNA binding of STAT3, thus blocking cell transformation [92]. In particular, drug discovery targeting STAT3 has been extensively undertaken in various diseases, and a large amount of evidence has supported the therapeutic potential of STAT3 inhibitors [93,94,95,96]. Nevertheless, to date, there have been no direct STAT inhibitors in clinics for the treatment of IBD. It has been reported that C188-9 has preventive effects in murine IBD models [22] (Table 1). Thus, the identification and development of phytochemicals targeting STAT proteins could have a significant role in the development of therapeutic drugs for IBD.

**Table 1 molecules-26-02824-t001:** Targeting the JAK–STAT pathway for IBD treatment.

	Compound	Target	Preclinical/Clinical Model	Dose/Daily	Ref.
JAK inhibitor	Tofacitinib	JAK1, JAK3	Approved	10, 20 mg	[97]
	Filgotinib	JAK1	PhaseII, III	200 mg	[98]
	Upadacitinib	JAK1	PhaseIII	24 mg	[99]
	Peficitinib	JAK1, JAK2, JAK3, TYK2	PhaseII	25, 75, 150 mg	[100]
	TD-1473	JAK1, JAK2, JAK3	PhaseII, III	20, 80, 270 mg	[52]
STAT inhibitor	C188-9	STAT3	DSS- or TNBS induced IBDmurine model	Not designated	[22]

## 3. Phytochemicals Targeting the JAK–STAT Pathway

Recently, phytochemicals have been highlighted as alternative/potent candidates for the management of IBD. Many studies have reported that plant-derived natural compounds are considered to have protective and therapeutic effects as dietary supplements for IBD [24]. It has also been suggested that phytochemicals can improve the intestinal barrier through various action mechanisms, including cytokine regulation and reduction of oxidative stress [25,101]. So far, it has been mainly focused on the ability of phytochemicals to downregulate the production of cytokines in IBD [23]. As discussed in the previous section, inhibition of JAK–STAT pathways prevents multiple pro-inflammatory cytokine signaling pathways, which can be considered a new therapy in IBD. Herein, we discuss the present evidence that phytochemicals could induce IBD remission by affecting the JAK–STAT pathway in animal model systems of IBD (Table 2). In addition, we analyzed human-relevant equivalent doses of each phytochemicals modulating JAK–STAT pathways, which suggests that these phytochemicals could be considered as candidates for new JAK–STAT inhibitors and have the potential for combinatorial use with current JAK inhibitors or other therapeutic drugs in IBD.

### 3.1. Phenolics

Phenolic compounds are secondary metabolites produced by plant cells, which possess a variety of bioactivities such as anti-inflammatory, antioxidant, antibacterial, and antiviral [102]. In particular, phenolic compounds have been well known to play anti-inflammatory roles by modulating diverse intracellular signaling pathways of inflammation and to have beneficial effects in various chronic inflammatory diseases [103].

#### 3.1.1. Curcumin

Curcumin is the most well studied phenolic compound derived from *Curcuma longa* and is known to have anti-mutagenic and anti-tumorigenic effects [104]. Several studies have shown that curcumin has anti-inflammatory effects against IBD. Accumulating evidence has revealed that curcumin has an anti-inflammatory effect on chronic inflammatory diseases by modulating JAK–STAT pathways [105]. In particular, some studies have shown that curcumin has beneficial effects in murine models of IBD. It reduces the severity of dextran sulfate sodium (DSS)-induced colitis by blocking the DNA binding of STAT3 and suppressing STAT3 phosphorylation in the mouse colon [106]. STAT3 has been reported to be highly phosphorylated in IBD patients and to be further involved in colitis-associated cancer as well as colonic inflammation [107]. Another study reported that curcumin can ameliorate trinitrobenzene sulfonic acid (TNBS)-induced severe colitis by downregulating the phosphorylation of JAK2, STAT3, and STAT6 in colonic tissues [108]. It was also found that pretreatment with curcumin in TNBS-induced IBD reduced inflammatory tissue damage by inhibiting the phosphorylation of STAT1 [108]. This suggests that curcumin could be used as a therapeutic intervention for human intestinal inflammation. Although further clinical studies are needed, such as precise dose of administration, curcumin could reduce clinical symptoms of IBD patients and be an effective therapy in humans [109].

#### 3.1.2. EGCG

Epigallocatechin-3-gallate (EGCG), a major bioactive polyphenol in green tea, is known to suppress inflammation and oxidative stress [110]. Many studies have shown that EGCG has anti-inflammatory effects on chronic inflammatory diseases, such as neurodegenerative diseases and cancers, in a multifactorial manner [111,112]. It improves acetic acid-induced colitis by reducing oxidative stress by decreasing nitric oxide (NO) production, increasing superoxide dismutase (SOD) expression, and inhibiting the production of TNF-α and IFN-γ in rats [113]. Treatment of DSS-induced colitis mice with EGCG ameliorated colitis by reducing malondialdehyde (MDA) caused by reactive oxygen species (ROS), which is one of the effector mechanisms of inflammation [114]. A recent study showed that EGCG downregulated cytokine IL-6 and reduced the expression of STAT3 protein in the colon tissue of colitis-induced mice. Therefore, EGCG reduced UC-like disease activity [115].

#### 3.1.3. Ellagic Acid

Ellagic acid, found in a wide range of fruits and vegetables, including *Punica granatum* (pomegranate), is known to have various biological activities [116]. In particular, many experimental studies have reported that ellagic acid has significant anti-inflammatory activities in the GI tract [117]. The ellagic acid and ellagic acid-rich fraction of pomegranate have antiulcerative effects in DSS-induced mice and rats [118,119]. Regarding specific anti-IBD molecular mechanisms, ellagic acid decreased inflammatory cytokines (IL-6, TNF-α, and IFN-γ) and crucial inflammatory mediators such as cyclooxygenase 2 (COX-2) and inducible nitric oxide synthase (iNOS), and blocked the expression and activation of STAT3 signaling pathways along with mitogen-activated protein kinase (MAPK) and nuclear factor kappa B (NF-κB), resulting in a decrease in disease severity in both acute and chronic colitis [120,121].

#### 3.1.4. Gallic Acid

Gallic acid, also known as 3,4,5-trihydroxybenzoic acid, is a naturally occurring phenolic compound found in fruits, nuts, and vegetables, and has been shown to have anti-inflammatory properties in a variety of chronic inflammatory disorders [122,123]. Gallic acid ameliorated UC-like clinical symptoms by reducing the phosphorylation of STAT3 and decreasing p65-NF-κB expression in the colon of DSS-induced mice [124]. Another study showed that gallic acid improved disease severity in DSS-induced mice by upregulating nuclear factor erythroid 2-related factor 2 (Nrf2) protein expression and downregulating the production of IL-21 and IL-23 [125].

#### 3.1.5. Paeonol

Paeonol, 2-hydroxy-4-methoxy acetophenone, is a major phenolic compound found in *Paeonia suffruticosa*, which has been used in traditional medicine and has been reported to have various bioactivities [126]. In particular, several studies have shown that paeonol exhibits anti-inflammatory effects in many inflammation-related diseases [127]. It reduced TNBS-induced colitis and suppressed IFN-γ-induced STAT1 activation in colon cancer-derived CW-2 cells and T cell leukemia-derived Jurkat cells [128]. Oral administration of paeonol in colitis animal models (mice or rats) reduced colitis symptoms, suggesting that paeonol could be a therapeutic intervention for the treatment of IBD [129,130].

#### 3.1.6. Piceatannol

Piceatannol is a hydroxylated derivative of resveratrol, which is a natural stilbene of phenolic compounds in grapes, berries, and passion fruits [131]. It is well known to have anti-inflammatory and suppressive effects on tumors [132,133]. Its anti-inflammatory activities target various inflammatory mediators such as iNOS, COX2, and NK-κB in vivo [134,135]. In DSS-induced colitis mice, piceatannol has been reported to ameliorate clinical signs in colonic tissue [136]. In this case, it exerted anti-inflammatory effects in the colon by decreasing the phosphorylation of STAT3, in addition to reducing the expression of iNOS, COX2, and crucial inflammatory cytokines such as TNF-α and IL-6.

#### 3.1.7. Shikonin

Shikonin is a major component extracted from the root of *Lithospermum erythrorhizon* and has been studied as a potential anticancer and anti-inflammatory drug [137]. The anti-inflammatory effect of shikonin was verified in several in vivo model systems, which attenuated the pathological symptoms by reducing inflammation by inhibiting NF-κB pathways [138,139,140]. Furthermore, shikonin reduced disease symptoms by blocking the activation of STAT3 and reducing colonic inflammatory cytokines, including IL-1β, IL-6, and TNF-α in a DSS-induced UC model [141]. This suggests that shikonin can be used as a therapeutic agent for treating IBD.

### 3.2. Terpenoid

Terpenoids are secondary metabolites widely distributed in plants and are known to have significant therapeutic potential against inflammatory diseases [142]. Many studies have verified that terpenoids exert anti-inflammatory effects by targeting various inflammatory mediators in in vivo inflammatory disease model systems [143].

#### Triptolide

Triptolide is a bioactive diterpenoid extracted from *Tripterygium wilfordii* [144]. It suppresses colitis symptoms by inhibiting the IL-6/STAT3 signaling pathway in IL-10 deficient mice as well as in an in vitro culture of colonic explants of patients with CD [145]. It also inhibits the progression from colitis to colon cancer in a 1, 2-dimethylhydrazine (DMH)/DSS-induced mouse model and downregulates the JAK–STAT3 pathway by the phosphorylation of STAT3 in colorectal cancer cells [146].

### 3.3. Nitrogen-Containing Alkaloids and Sulfur-Containing Compounds

#### 3.3.1. Boldine

Boldine, an aporphine alkaloid found in the boldo tree, has been used as a traditional remedy in several diseases and is well known for its anti-tumor, anti-atherogenic, and anti-diabetic effects [147,148,149]. In terms of anti-inflammatory activity, boldin has been reported to attenuate DSS-induced colon damage in mice by inhibiting inflammatory processes by increasing antioxidant enzymes SOD and catalase (CAT) and decreasing the expression and activation of STAT3 protein as well as p65 NF-κB in the colon [150].

#### 3.3.2. Berberine

Berberine, a benzylisoquinoline alkaloid found in the *Berberis* species, has been reported to have beneficial effects on DSS- or TNBS-induced colitis mouse models through various molecular mechanisms [151]. Berberine showed anti-colitic activity in TNBS-treated mice by blocking the expansion of Th1 and T helper type 17 (Th17) cells by reducing the phosphorylation of STAT1 and STAT3 in CD4+ T cells isolated from mice [152]. In addition, it ameliorated clinical severity in DSS-induced chronic relapsing mice by inhibiting the differentiation of Th17 cells by downregulating the phosphorylation of STAT3 [153]. Another study reported that berberine is involved in the protection of the intestinal barrier, not only by reducing pro-inflammatory cytokines and oxidative stress mediators such as myeloperoxidase (MPO), but also by increasing the expression of tight junction proteins such as zonula occludin-1 (ZO-1), resulting in relieving colitic symptoms in DSS-induced colitis mice [154]. In addition, a recent study demonstrated that berberine inhibited the phosphorylation of JAK1/2 and STAT1/3/4/5/6 through oncostatin M, belonging to the IL-6 cytokine family, leading to the attenuation of gut inflammation in DSS-induced mice [155].

#### 3.3.3. Garlic Organosulfur Compounds (Allicin, Diallyl Trisulfide, and Alliin)

Garlic has long been used as food or traditional medicine for many diseases, and many studies have verified through in vitro and in vivo model systems that garlic organosulfur compounds have various biological activities, including anti-inflammatory, anti-oxidative, anti-diabetic, and anticancer properties [156]. Garlic is composed of several major organosulfur compounds, some of which have been reported to have anti-inflammatory activity in vivo [157].

Allicin is a sulfur-containing natural compound obtained from garlic, *Allium sativum*, which has various biological activities [158]. In DSS-induced colitis mice, allicin reduced colon damage by suppressing the phosphorylation of STAT3 and inhibiting the expression of NF-κB [159].

Diallyl trisulfide (DATS) is an organic polysulfide from garlic that has been reported to ameliorate disease symptoms in DSS-induced colitis mice by blocking the DNA binding and phosphorylation of STAT3 [160].

Alliin is a well-known sulfur-containing compound in garlic. Treatment with alliin in DSS-induced colitis mice reduced colon damage, showing a decrease in pro-inflammatory cytokines (IL-6, IL-1β, and TNF-α) in colonic tissue [161]. In addition, alliin downregulated the activation of STAT1, MAPK, and NF-κB in lipopolysaccharide (LPS)-stimulated macrophages.

#### 3.3.4. Phenethylisothiocyanate

Phenethylisothiocyanate (PEITC) is enriched in many cruciferous vegetables and is well known for its anti-cancer activity [162]. Relatively few studies have investigated the anti-inflammatory effects of PEITC. PEITC has been reported to alleviate histopathological signs in the acute and chronic inflammatory colon induced by DSS and attenuate the activation of STAT1 in LPS-activated macrophages. Therefore, blocking the activation of STAT1 by PEITC is associated with a reduction in IBD [163].

**Table 2 molecules-26-02824-t002:** Phytochemicals targeting JAK–STAT pathways in inflammatory bowel disease models.

Class of Phytochemicals	Phytochemical Name	Experimental System	Effective Doses(mg/kg Body Weight, Daily)	Translated into Human-Relevant Equivalent (mg/kg)	Target of JAK–STAT Pathway	Main Source	Ref.
Phenolic	Curcumin 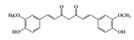	DSS-inducedTNBS-induced	36.8, 92100	2.9, 7.48.1	JAK2, STAT1, 3, 6	*Curcuma longa*Linn (turmeric)	[106][108]
	EGCG 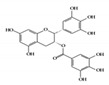	DSS-induced	50, 100	4.0, 8.1	STAT3	green tea	[115]
	Ellagic acid 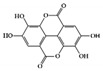	DSS-induced	100	8.1	STAT3	Pomegranate (*Punica granatum* L., *Lythraceae*)	[120]
	Gallic acid 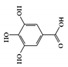	DSS-induced	10	0.8	STAT3	Green tea, strawberries, grapes, bananas, and many other fruits	[124]
	Paeonol 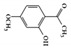	TNBS-induced	0.5 mg/kg treated intrarectally	0.04	STAT1	Moutan Cortex	[128]
	Piceatannol 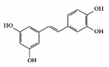	DSS-induced	10	0.8	STAT3	Grapes, rheum undulatum, rhubarb, and sugar cane	[136]
	Shikonin 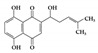	DSS-induced	25	2.0	STAT3	*Lithospermum erythrorhizon*	[141]
Terpenoid	Triptolide 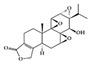	IL-10 deficient colitis mice	0.07 mg/kg treated intraperitoneally	0.005	STAT3	*Tripterygium Wilfordii* Hook. f	[146]
Nitrogen containing alkaloid	Boldin 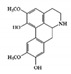	DSS-induced	50	4.0	STAT3	Boldo tree	[150]
	Berberine 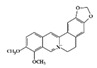	DSS-induced	2050	1.64.0	STAT3JAK1, 2 and STAT1, 3, 4, 5, 6	*Berberis* species	[153][155]
Organosulfur compounds	Allicin 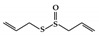	DSS-induced	10	0.8	STAT3	*Garlic*	[159]
	Diallyl trisulfide 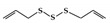	DSS-induced	45, 90	3.6, 7.2	STAT3	*Garlic*	[160]
	Alliin 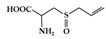	DSS-induced	500	40.5	STAT1	*Garlic*	[161]
	Phenethylisothiocyanate (PEITC) 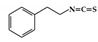	DSS-induced	75	6.0	STAT1	cruciferous vegetables	[163]

All phytochemicals orally administered, except paeonol (intrarectal administration) and triptolide (intraperitoneal administration).

## 4. Challenges of Phytochemicals Targeting JAK–STAT Pathways for IBD Treatment

In this section, we discuss our current knowledge and significance of phytochemicals as natural resources in the discovery of JAK inhibitors or STAT inhibitors. A total of 14 phytochemicals have been confirmed to have anti-IBD effects by inhibiting JAK, STAT, or both in in vivo murine model systems (Figure 3). Phytochemicals modulating JAK–STAT pathways can have several strengths as potential inhibitors against IBD. Currently, developing or developed JAK inhibitors have been reported to have various adverse effects, including higher rates of influenza, herpes zoster infection, arthralgia, and headache [55,97,164]. In this regard, phytochemicals targeting the JAK–STAT pathway having fewer side effects could be advantageous for IBD treatment [26]. In addition, phytochemicals have been found to have a wide array of anti-inflammatory activities in IBD [165]. Indeed, as shown in Section 3, the majority of phytochemicals have been reported to simultaneously have other anti-inflammatory effects together with modulation of the JAK–STAT pathway. Accordingly, phytochemicals might be able to provide resources for new drugs targeting JAK–STAT pathways and could be an alternative and complementary medicine for treating IBD patients. In addition, there could be combinatorial availability of phytochemicals along with current therapies for IBD. A combination of diverse biological agents has been typically used in IBD, but clinical data for safety and efficacy have been limited [166]. Although there are no current data on the combination of phytochemicals targeting JAK–STAT pathways and existing IBD therapies, they could have attractive potential for IBD treatment because of the fewer side effects of phytochemicals. Furthermore, intestinal enteroid/organoid culture from intestinal stem cells (ISCs) have been recently considered as new therapeutic strategies for the treatment of refractory IBD [167,168]. Transplantation of intestinal organoid in IBD patients could be used to promote the regeneration of the damaged intestinal tissue and also provide new scaffolding for the discovery of novel drugs for IBD treatment. Therefore, anti-colitic activity of phytochemicals targeting JAK–STAT pathways could be investigated in intestinal organoid cultures mimicking in vivo physiology of IBD. In addition, the combination of these phytochemicals and the transplantation of intestinal organoid could be the advanced strategy for IBD treatment.

Although phytochemicals have potential for developing new anti-JAK–STAT drugs for IBD, there are some obstacles. Phytochemicals targeting JAK–STAT signaling in the previous section have only been shown to ameliorate disease symptoms in experimental rodent models. Thus, the safety and efficacy identified in animal studies must be translated into human trials. In this regard, we summarized anti-colitic phytochemicals based on structural classification [169,170] and analyzed the effective doses in each experimental system (Table 2). A large portion of effective compounds are phenolic compounds, and the second most dominant class is organosulfur compounds. Furthermore, we translated the effective doses of phytochemicals in rodent model systems into human-relevant equivalent doses [171]. Given the orally administered phytochemicals, translated doses of these phytochemicals ranged from a minimum of 0.8 mg/kg to a maximum of 40.5 mg/kg. The most effective phytochemicals against IBD are gallic acid, piceatannol, and allicin, with a human-relevant equivalent dose of 0.8 mg/kg. These three phytochemicals seem to have anti-colitic activity at a much lesser dose than the effective dose of JAK inhibitors (filgotinib: 200 mg once daily → 3.3 mg/kg in case body mass weighs 60 kg) that have been used for treating IBD patients as shown in Table 1. In addition, as shown in Table 2, a majority of phytochemicals targeting JAK–STAT pathways seem to be involved in the inhibition of STAT3 activity. STAT3 has been considered the most crucial factor among STAT family members in colitis-associated colorectal cancer due to its crosstalk with cytokine-mediated chronic inflammation in the intestine [172]. Although several small-molecule STAT3 inhibitors have been clinically investigated for treating advanced solid tumors, including colorectal cancer, and have also been preclinically studied in murine arthritis and athema models [93,94,96], there have been no direct STAT inhibitors for clinical application against IBD. Thus, phytochemicals modulating STAT3 could be a good resource for the discovery of new STAT3 inhibitors for the treatment of IBD.

A large number of studies have shown that functional nutraceuticals can reduce inflammation of the GI tract by regulating proinflammatory cytokines in IBD [173]. Nutraceutical combines two words, ‘nutrient’ and ‘pharmaceutical’. Phytochemicals have been considered nutraceutical agents against autoimmune disorders [174]. In this respect, phytochemicals targeting JAK–STAT pathways could be significant resources for developing new nutraceutical agents with potentially strong medicinal properties against IBD.

## 5. Conclusions and Perspective

JAK–STAT-mediated cellular signaling responses play an essential role in both enteric homeostasis and IBD pathology, which is characterized by uncontrolled chronic inflammation in the GI tract. Numerous inflammatory cytokines are chronically produced and released in the GI tract and regulate the JAK–STAT signaling pathway in IBD. In addition to targeting particular cytokines such as TNF-α, the clinical significance of the JAK–STAT pathway in IBD has been emphasized by an ongoing preclinical and clinical study using JAK and STAT inhibitors for the treatment of IBD patients. Targeting the JAK–STAT pathway can prevent multiple pro-inflammatory cytokine signaling pathways in IBD. The orally available JAK inhibitor, tofacitinib, approved by the FDA, has been clinically confirmed to have beneficial effects in patients with moderate to severe UC, but considering safety issues, tofacitinib-treated patients have shown various adverse effects, including a high rate of infection [175]. It is imperative to consider phytochemicals that can modulate JAK–STAT pathways as potential candidates for IBD treatment. In this respect, this review has focused on particular phytochemicals that inhibit JAK–STAT pathways and suppress IBD pathology in vivo in animal model systems, among many phytochemicals with anti-inflammatory effects against IBD. The phytochemicals are shown in Table 2 have been shown to protect against colonic inflammation and ameliorate disease symptoms by downregulating JAK–STAT signaling in murine IBD models. The effective dose of phytochemicals in the IBD murine model was translated into a human-relevant equivalent dose, and the phytochemicals were considered to have competitive power as potential candidates for the treatment of IBD. However, more studies focusing on the precise mechanism of suppressing JAK–STAT pathways are required to understand the relationship between its structural characteristics and anti-JAK–STAT activity or cell type specificity of each phytochemical in IBD. Furthermore, to verify the beneficial effects of these phytochemicals in managing IBD, their therapeutic utilization requires safety and efficacy data through both pre-clinical and clinical studies in IBD. Taken together, we provide a fundamental understanding of phytochemicals modulating JAK–STAT pathways to develop new therapeutic drugs, alternative medicines, and nutraceutical agents for treating IBD.

## Figures and Tables

**Figure 1 molecules-26-02824-f001:**
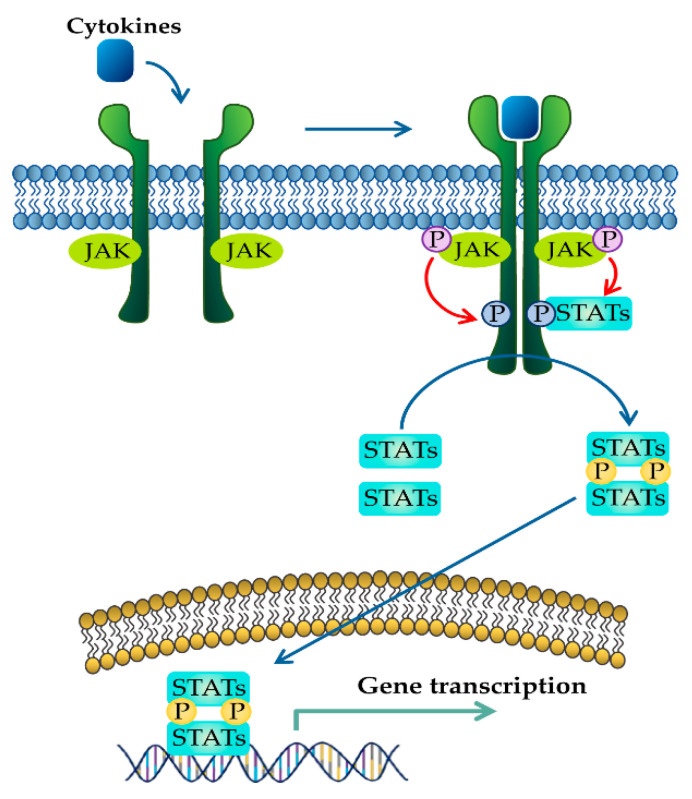
JAK–STAT signaling pathway activated in response to cytokines. Binding of cytokines to their cognate receptors triggers the phosphorylation of JAK and its receptors. After that, recruited STAT is phosphorylated and translocated as homo- or heterodimers to the nucleus, where they upregulate the transcription of cytokine-responsive genes.

**Figure 2 molecules-26-02824-f002:**
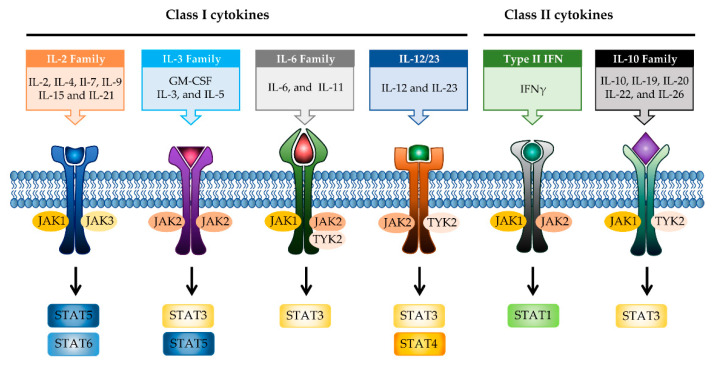
JAK-STAT signaling pathways in IBD. Multiple combinations with JAK proteins and STAT proteins are determined depending on the cytokines and their cognate receptors. Each cytokine family playing a key role in IBD pathogenesis is divided into two classes. JAK, janus kinase; STAT, signal transducer activator and activation of transcription; IL, interleukin; GM-CSF, granulocyte-macrophage colony-stimulating factor; IFN, interferon.

**Figure 3 molecules-26-02824-f003:**
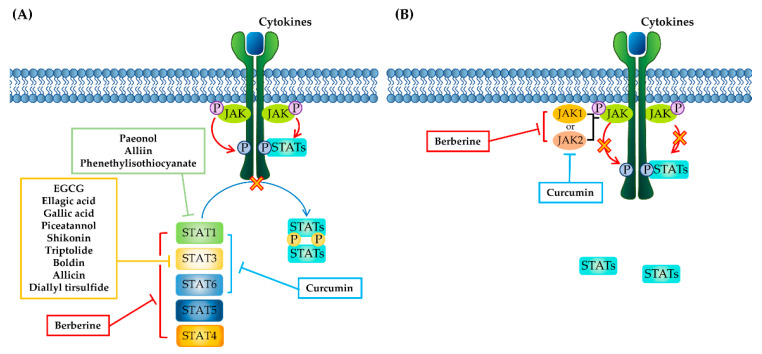
Phytochemicals targeting JAK–STAT pathways. (**A**) Phytochemicals targeting STAT proteins. Phytochemicals in green box (paeonol, alliin, and phenethylisothiocyanate) inhibit the activation of STAT1. Phytochemicals in yellow box (EGCG, ellagic acid, gallic acid, piceatannol, shikonin, triptolide, boldin, allicin, and diallyl trisulfide) inhibit the activation of STAT3. Berberine inhibits the phosphorylation of STAT1/3/4/5/6. Curcumin inhibits the activation of STAT1/3/6. (**B**) Phytochemicals targeting JAK proteins. Berberine inhibits the phosphorylation of JAK1/2. Curcumin downregulates the phosphorylation of JAK2.

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
