# Peer review of "Phytochemicals Targeting JAK–STAT Pathways in Inflammatory Bowel Disease: Insights from Animal Models"

_molecules, 2021, doi:10.3390/molecules26092824_

Round 1

Reviewer 1 Report

The review summarizes current studies on phytochemicals targeting the JAK-STAT pathway in inflammatory bowel disease (IBD). In general, the review is well written and logically organized with a proper citation of references.  

Author Response

<Reviewer 1>

Thank you very much for giving us an opportunity to submit a revised manuscript. Our responses to the reviewers’ comments are included in this letter. In addition, we have modified some parts of the manuscript in response to the reviewers’ comments. Modified parts are indicated with yellow color. We hope that you find the revised manuscript acceptable for publication in ‘Molecules’.

Comments and Suggestions for Authors

The review summarizes current studies on phytochemicals targeting the JAK-STAT pathway in inflammatory bowel disease (IBD). In general, the review is well written and logically organized with a proper citation of references.

Thank you very much for your positive response!

<Reviewer 2>

Thank you very much for giving us an opportunity to submit a revised manuscript. Our responses to the reviewers’ comments are included in this letter. In addition, we have modified some parts of the manuscript in response to the reviewers’ comments. Modified parts are indicated with yellow color. We hope that you find the revised manuscript acceptable for publication in ‘Molecules’.

Comments and Suggestions for Authors

The Authors, in a systematic and easy-to-read manner, adequately structured this review by focusing on the main inflammatory mechanisms involved in the development of IBD and the potential inhibitory role of numerous nutritional molecules.

Although this review is very well structured, the Authors could improve their manuscript by adding the strategy carried out through a PubMed literature search to gather the available evidences (such as: gathered through a PubMed literature search, from - to, including the following terms:).

Per this suggestion, we added the strategy for PubMed literature search in section 1. Introduction. (line 72~75)

- line 72-75

“To investigate the function of phytochemicals targeting JAK-STAT pathways in IBD, we searched for articles in PubMed with three main key words (phytochemical, IBD, and anti-inflammation). In addition, we limited phytochemicals that regulated JAK-STAT pathways in in vivo animal model systems.”

Moreover, considering the innovative in vitro models available in this field of research (Organoid-based regenerative medicine for inflammatory bowel disease. Okamoto R et al Regen Ther 2020 Jan 13;13:1-6. Intestinal enteroids/organoids: A novel platform for drug discovery in inflammatory bowel diseases; Yoo JH, Donowitz M. World J Gastroenterol. 2019 Aug 14;25(30):4125-4147) it could be very useful to add a small paragraph dedicated to this topic.

Per this suggestion, we added the small paragraph in section 4. Challenges of phytochemicals targeting JAK-STAT pathways for IBD treatment. (line 428~436)

- line 428~436

“Furthermore, intestinal enteroid/organoid culture from intestinal stem cells (ISCs) have been recently considered as new therapeutic strategies for the treatment of refractory IBD [167,168]. Transplantation of intestinal organoid in IBD patients could be used to promote the regeneration of the damaged intestinal tissue and also provide new scaffolding for the discovery of novel drugs for IBD treatment. Therefore, anti-colitic activity of phytochemicals targeting JAK-STAT pathways could be investigated in intestinal organoid cultures mimicking in vivo physiology of IBD. In addition, the combination of these phytochemicals and the transplantation of intestinal organoid could be the advanced strategy for IBD treatment”

Minor:

  1. Please, add citation of Figure 1 in lines 57/58

Per this comment, we added the citation of Figure 1 in line 59.

  1. Various acronyms (TNFa, INFa, COX2 and iNOS) should be written in full

Per this suggestion, we rewrote various acronyms in full. When we used the full terms at the first time, we put various acronyms in parentheses after it. Thereafter, we used the acronyms, instead of full names.

<Reviewer 3>

Thank you very much for giving us an opportunity to submit a revised manuscript. Our responses to the reviewers’ comments are included in this letter. In addition, we have modified some parts of the manuscript in response to the reviewers’ comments. Modified parts are indicated with yellow color. We hope that you find the revised manuscript acceptable for publication in ‘Molecules’.

Comments and Suggestions for Authors

This article is very well reviewed JAK-STAT pathways involved in inflammatory bowel diseases and some efficacious medicines, such as antibody drugs and phytochemicals which inhibit JAK-STAT pathways. However, the following points would be considered to improve this article.

Minor points

1) Line : 241

section, JAK-STAT pathways prevent multiple pro-inflammatory cytokine signaling

Reviewer wonder if this content is opposite. Inhibition of JAK-STAT pathways prevent multiple pro-inflammatory cytokine signaling

Thank you very much for your comment! We corrected the wrong sentence. (JAK-STAT pathways prevent multiple pro-inflammatory cytokine signaling → inhibition of JAK-STAT pathways prevents multiple pro-inflammatory cytokine signaling pathways (line 246))

2) Line : 266

Trinitrobenzene sulfonic acid (TNBS). The character size is different from other words.

Per this comment, we corrected the character size of “Trinitrobenzene sulfonic acid (TNBS)” (10.5→10) (line 271)

Reviewer 2 Report

The Authors, in a systematic  and easy-to-read manner, adequately structured this review by focusing on the main inflammatory mechanisms involved in the development of IBD and the potential inhibitory role of numerous nutritional molecules.

Although this review is very well structured, the Authors could improve their manuscript by adding the strategy carried out through a PubMed literature search to gather the available evidences (such as: gathered through a PubMed literature search, from - to, including the following terms:  ).

Moreover, considering the innovative in vitro models available in this field of research (Organoid-based regenerative medicine for inflammatory bowel disease. Okamoto R et al Regen Ther 2020 Jan 13;13:1-6. Intestinal enteroids/organoids: A novel platform for drug discovery in inflammatory bowel diseases;  Yoo JH, Donowitz M. World J Gastroenterol. 2019 Aug 14;25(30):4125-4147) it could be very useful to add a small paragraph dedicated to this topic.

Minor:

  1. Please, add citation of Figure 1 in lines 57/58
  2. Various acronyms (TNFa, INFa, COX2 and iNOS) should be written in full

Author Response

(The authors gave the same response as above.)

Reviewer 3 Report

This article is very well reviewed JAK-STAT pathways involved in inflammatory bowel diseases and some efficacious medicines, such as antibody drugs and phytochemicals which inhibit JAK-STAT pathways.

However, the following points would be considered to improve this article.

Minor points

1) Line : 241

section, JAK-STAT pathways prevent multiple pro-inflammatory cytokine signaling

Reviewer wonder if this content is opposite.

Inhibition of JAK-STAT pathways prevent multiple pro-inflammatory cytokine signaling

2) Line : 266

Trinitrobenzene sulfonic acid (TNBS)

The character size is different from other words.

Author Response

(The authors gave the same response as above.)
